# Influence of Hydroxyapatite Nanoparticles and Surface Plasma Treatment on Bioactivity of Polycaprolactone Nanofibers

**DOI:** 10.3390/polym12091877

**Published:** 2020-08-20

**Authors:** Eva Stastna, Klara Castkova, Jozef Rahel

**Affiliations:** 1CEITEC–Central European Institute of Technology, Brno University of Technology, Purkynova 123, 61200 Brno, Czech Republic; castkova@fme.vutbr.cz; 2Faculty of Mechanical Engineering, Brno University of Technology, Technicka 2, 61269 Brno, Czech Republic; 3Faculty of Science, Department of Physical Electronics, Masaryk University, Kotlarska 2, 61137 Brno, Czech Republic

**Keywords:** electrospinning, nanofibers, polycaprolactone, hydroxyapatite, tissue engineering, atmospheric pressure argon plasma

## Abstract

Nanofibers are well known as a beneficial type of structure for tissue engineering. As a result of the high acquisition cost of the natural polymers and their environmentally problematic treatment (toxic dissolution agents), artificial polymers seem to be the better choice for medical use. In the present study, polycaprolactone nano-sized fibrous structures were prepared by the electrospinning method. The impact of material morphology (random or parallelly oriented fibers versus continuous layer) and the presence of a fraction of hydroxyapatite nanoparticles on cell proliferation was tested. In addition, the effect of improving the material wettability by a low temperature argon discharge plasma treatment was evaluated, too. We have shown that both hydroxyapatite particles as well as plasma surface treatment are beneficial for the cell proliferation. The significant impact of both influences was evident during the first 48 h of the test: the hydroxyapatite particles in polycaprolactone fibers accelerated the proliferation by 10% compared to the control, and the plasma-treated ones enhanced proliferation by 30%.

## 1. Introduction

Modern medicine still struggles with treating certain types of large-scale injuries such as bone defects caused by accidents, bone tumors, or vast skin burns. For bone defects, bio-inert materials are usually used for the treatment, typically titanium, zirconia, or alumina. However, some problems are related to these materials. The different mechanical properties of these materials (e.g., hardness, stiffness) induce changes of mechanical properties in the surrounding bone [1,2]. For these reasons, modern types of implants are searched for. An ideal implant material has similar mechanical properties as the natural tissue, provides mechanical support for the cells rebuilding the damaged tissue, induces the activity of the living cells, and the body is able to absorb it gradually during the remodeling process. It is obvious from these requirements that the chemical composition must be similar to that of the natural tissue. Furthermore, its structure must be penetrable by the cells, and its dissolution rate must be comparable with the tissue remodeling time. Therefore, various types of polymers and ceramics are widely explored for use in tissue engineering. To mimic the natural structure of body tissue, micro- and nanofibrous scaffold structures seem to be promising [3,4,5,6].

There are several methods of micro- and nanofiber preparation: electrospinning, force spinning, phase separation, self-assembly etc. [3,4,7]. The electrospinning method uses the effect of a high electric field on an electrically conductive liquid-phase precursor to form ultrafine fibers having a diameter of approximately 100–1000 nm. It is a versatile process, and by the control of the process parameters, it is possible to change the morphology of the obtained fibers (e.g., diameter of the fibers and their directional alignment). By careful control of the electrospinning precursor’s chemical composition, it is also possible to fabricate diverse types of composite structures. The main disadvantage of the process is its sensitivity to small changes in process parameters and external ambient conditions (e.g., air humidity and temperature) [5,8,9].

Natural polymers (collagen, chitosan, polylactic acid, etc.) seem to be a logical choice for fibrous scaffolds since they are naturally present in the body. However, they are often soluble only in toxic solvents (e.g., collagen needs fluoride ions for the dissolution), the solutions are not stable for a long time, and the price of the natural polymers is high [5,7,10,11]. On the other hand, artificial polymers are not only cheaper (e.g., polycaprolactone, polyvinyl alcohol), but their solutions can be prepared with lower health and environmental risk. Several previous studies have already proved the biocompatibility and bioactivity of polycaprolactone and polyvinyl alcohol [6,7,12]. The importance of structure morphology (e.g., porosity, directional alignment) and surface treatment (functionalization) on the bioactivity was addressed in several works, too [13,14,15].

Polycaprolactone (PCL) is a versatile biocompatible polymer that can be easily processed by various methods into PCL fibers, films, and microporous bulk. The biocompatibility was typically tested in vitro by the direct contact method with various types of cells (fibroblasts, keratinocytes, mesenchymal stem cells, etc.) [16,17,18,19]. Some in vivo tests were also conducted e.g., on macroporous samples prepared by selective laser sintering. The tests proved the successful formation of a new bone tissue and its mineralization [17].

Electrospun PCL fibers were investigated as a potential scaffold for bone tissue regeneration, too. In the work of Ajinkya Shitole et al. [20], the PCL fibers doped with hydroxyapatite (HA) and HA + ZnO nanoparticles were presented. The hydroxyapatite—a bioceramic with a chemical composition similar to that of natural bone—was intended to support the scaffold bioactivity, and the ZnO nanoparticles should provide an antibacterial effect. The cell proliferation of PCL/HA samples was better than reference pure PCL even after 7 days of testing. In addition to that, HA particles increased the hydrophilicity, which is a plausible reason for increased cell activity. The positive effect of the hydroxyapatite particles on bioactivity was also mentioned in the work of Fattane Keivani et al. [21] and Izabella Rajzer et al. [22].

In vivo tests of PCL and PCL/HA composite fibers were done by Andrade et al. [23]. However, their fibrous structure was prepared by a rotary jet-spinning, not by electrospinning, which means that the resulting fibers had higher diameter (up to 1.9 µm) than typical electrospun fibers (approximately 100 nm). The fibrous scaffolds were tested on an artificial bone defect of cylindrical shape with a diameter of 3.0 mm. The scaffolds were both biocompatible and bioactive. Increasing the amount of the hydroxyapatite particles incorporated into the fibers increased the rate of new mineralized matrix formation [23].

The surface properties of polymer fibers can be modified by various surface treatments. The first possibility is their bioactive and hydrophilic coating. The typical coating consists of collagen or another natural polymer or substance—for example, poly(dopamine). The coating strongly supports the bioactivity and incorporation of the scaffold to the surrounding tissue [24,25]. However, there is a problem with the even thickness of the coating and with the clogging of inter-fibrous space by the coating material. The second approach relies on increasing the fiber surface wettability only. This can be relatively easily facilitated e.g., by plasma treatment (activation). In general, plasma activation introduces a small amount of polar functional groups to the polymer surface, which contributes to its improved wettability via the increased surface free energy of the polymer. The improved wettability is known to support the cells’ activity [26,27,28]. This is important especially for materials with microfibrous topology, where incomplete wetting of such a heterogeneous surface may result in the well-known lotus effect [29]. The main functional principle behind the plasma activation is the formation of polar functional groups (typically –COOH or –C=O), which are strongly interacting with dipoles of surrounding water molecules [30,31,32]. Formation of these groups is observed not only in oxygen-containing but also in inert gas plasmas (i.e., Ar, He, N_2_). There, the formation of oxygen-containing polar groups can be attributed to the fission of the adsorbed gas molecules on the fibers’ surface (e.g., H_2_O, O_2_, CO_2_) by the flux of plasma-generated energetic particles (UV, electrons, ions, metastables, etc.). The benefit of using inert gas plasmas lays in its lesser tendency to induce excessive oxidative degradation of the treated polymer.

Multi-phase nanofibrous systems as well as plasma treatment are both capable of promoting the bioactivity of promising scaffold materials. However, so far, not much research work has been done on the combination of both factors. The synergic effect of the ceramic phase in the polymer nanofibers and the surface treatment of the nanofibers promises an improvement of the biological properties of the structure that is beneficial for bone tissue engineering. The presented paper reports on the synergic effect of low-temperature plasma surface treatment applied on the polymer nanofibers with a hydroxyapatite phase incorporated within its fibrous structure. The in vitro testing of the prepared materials was carried out, and the effect on the biological properties is described in detail.

## 2. Materials and Methods 

### 2.1. Precursor Preparation and Analysis

Two precursors were prepared for the electrospinning. The first solution served as a precursor for the electrospinning of polymer fibers. Polycaprolactone (Mw = 80,000 g/mol, Sigma Aldrich, Steinheim, Germany) was dissolved in a mixture of acetic acid (glacial, Merck, Darmstadt, Germany) and formic acid (p.a., Penta, Praha, Czech Republic) in a 3:1 weight ratio (polycaprolactone:acetic acid) to obtain a 10 wt % solution of the polycaprolactone. The precursor was mixed for five hours before the electrospinning. 

The second precursor served for the preparation of ceramic–polymer composite nanofibers. Hydroxyapatite nanoparticles were prepared by the co-precipitation method of calcium phosphate (Lach-ner, Neratovice, Czech Republic) and diammonium hydrogenphosphate (Lach-ner, Czech Republic) in a stoichiometric ratio. The first solution was prepared by the dissolution of calcium nitrate hexahydrate in distilled water to obtain 14.5 wt % solution. The basicity of the solution was modified by adding ammonium hydroxide (Lach-ner, Czech Republic) to pH = 10. The second solution was 6 wt % solution of diammonium hydrogenphosphate in distilled water. Continuously stirred at room temperature, the diammonium hydrogenphosphate solution was drip added into the calcium nitrate solution. Then, the resulting suspension was stirred for one hour. Then, the suspension was hydrothermally treated at 200 °C for 5 h (described in detail in [33]). After the treatment, the suspension was washed in distilled water to reach neutral pH and finally placed in isopropyl alcohol. The obtained powder was characterized by X-Ray diffraction analysis (XRD, SmartLab 3 kW, Rigaku, Tokyo, Japan), the morphology of the particles was analyzed by scanning electron microscope (SEM, Verios, FEI, Brno, Czech Republic), and the chemical composition of the particles was analyzed using energy-dispersive X-ray spectroscopy (EDX, Verios, FEI, Czech Republic) 

The hydroxyapatite particles were dispersed in the solution of acetic and formic acid in a weight ratio of 3:1 (acetic acid:formic acid). The suspension was dispersed using an ultrasound probe Sonopuls 2450 (Bandelin, Berlin, Germany) for 3 min, and then polycaprolactone was added to the dispersion. The final concentration of the constituents was 5 wt % of hydroxyapatite particles and 6 wt % of polycaprolactone. The precursor was stirred for five hours and then electrospun.

The dynamic viscosity of the clear polycaprolactone and polycaprolactone/hydroxyapatite precursor was measured on the rotational rheometer (HAKE MARS II, Thermo Scientific, Karlsruhe, Germany) equipped with parallel plate geometry. The measurement was provided at the temperature of 25 °C in the range of shear rate from 1000 to 1 Hz after 5 h from the preparation. 

### 2.2. Electrospinning

4Spin equipment (Contipro, Dolni Dobrouc, Czech Republic) was used for the electrospinning. Both precursors were electrospun under the same parameters listed in Table 1. The only parameter that varied was the rotation speed of fibers collector: randomly organized fibers were collected with the collector rotation speed of 300 rpm; for parallel fibers, the collector speed was set to 2000 rpm. A list summarizing the prepared samples and their marking is shown in Table 2.

To assess the effect of fibrous morphology on the biological performance, samples of continuous layers were prepared as well. The glass plate was washed in ethanol and dried. Then, a drop of the polycaprolactone or polycaprolactone/hydroxyapatite precursor was put on the glass plate and then spread to a uniform layer.

For structural analysis, nanofibers were collected on aluminum foil for 30 min. The fibers were then observed on SEM (Verios, FEI, Czech Republic) for evaluation of their morphology and average fiber diameter. The measurement was done on at least two SEM images to obtain at least 100 values. 

For bioactivity tests, the nanofibers were collected on thin glass plates with a diameter of 12 mm that were placed on the collector using duct tape. The samples were electrospun for 1 min only. The short electrospinning time ensured sample transparency for further in vitro analysis.

### 2.3. Surface Plasma Treatment

The fibers were plasma treated after the electrospinning. Plasma treatment was done by diffuse coplanar surface barrier discharge (DCSBD) [34], operating in flowing argon (3 sccm) at the pressure of one atmosphere. This particular way of plasma treatment was found to be thermally compatible with the low melting temperature of polycaprolactone (60 °C), while still offering a relatively short treatment time for achieving complete sample wetting. The DCSBD electrode system consisted of a pair of coplanar comb-like energizing electrodes with linear strips of 1.5 mm width and 1 mm lateral separation, which were screen-printed on the transformer oil insulated face of a 0.6 mm thick 96% Al_2_O_3_ dielectric plate. Upon energizing by 15 kHz sinusoidal high voltage at an input power of 90 W, a thin 0.3 mm layer of argon plasma was generated over the Al_2_O_3_ dielectric plate, covering an area of 8 × 20 cm^2^. The treated samples were fixed at the height of approximately 0.1 mm above the discharge electrode by a fine PET mesh (SEFAR PET 1500, 32/83-70 PW) inserted between the sample and the discharge electrode (Figure 1). All samples were treated for 30 s. The discharge plasma serves mainly as a source of intense energetic flux stored in Ar energized states (e.g., Ar(^4^S[^3^P_2_]), Ar(^4^P), Ar^+^ or Ar_2_^*^). These initiate a cascade of dissociative reactions with molecules of ambient gas and water adsorbed on the fibers’ surface, leading to the formation of polar hydrophilicity-improving functional groups of the fibers’ surface. Besides the samples listed in Table 2, sole hydroxyapatite particles were plasma-treated, too.

Chemical changes of the fibrous structures after the plasma treatment were analyzed by XPS analysis using AXIS SupraTM (Kratos Analytical, Manchester, UK). The analysis before and after the plasma treatment was done on samples PCL-2 (pure random polycaprolactone fibers) and on pure hydroxyapatite particles to resolve the influence of the surface treatment on both structural components.

The wetting properties of the investigated samples were evaluated by contact angle measurement using a sessile drop method. High-purity water was used as the testing liquid. The contact angle of a small drop (1 µL) of water was measured 2 s after each drop, using a portable computer-based instrument (SEE System, Advex Instrument, Brno, Czech Republic). The contact angle was determined as the average of 10 measurements.

### 2.4. Biological Properties

Samples listed in Table 2 were prepared for biocompatibility tests based on international standard EN ISO 10993-5. The cells from the L9292 line (mouse fibroblasts) were cultivated on the sample and were observed for 3 days under a confocal microscope (Zeiss, Oberkochen, Germany). A snapshot was made every two minutes, and a video record was taken from at least six fields of view. The number of cells was counted from the pictures taken after 30 min, 24, 48, and 72 h. Then, the specific growth rate (SGR) was calculated. The results were always compared to a control sample: a transparent glass plate.

## 3. Results and Discussion

### 3.1. Precursors Characterization

The synthesized hydroxyapatite particles had a needle-like shape with a diameter of approximately 30 nm and length of 500 nm. The SEM image and the XRD spectrum are shown in Figure 2a and Figure 2b, respectively. The XRD analysis confirmed the phase composition of the hydroxyapatite particles (characteristic peaks for hydroxyapatite are marked with symbol *). Broad peaks of the XRD spectrum are caused by the nanometer dimensions of the particles. The stoichiometric ratio of Ca/P = 1.67 typical for the hydroxyapatite was confirmed by the EDX analysis.

The results of the viscosity measurements are shown in Figure 3 and Figure 4. The pure polycaprolactone precursor showed an apparent hysteresis of dynamic viscosity (η) as well as inner shear stress (τ). The polycaprolactone/hydroxyapatite precursor did not show such distinct hysteresis of the shear stress. This phenomenon was attributed to the self-organization of polymer chains while increasing the shear rate. This alignment disappeared when the shear rate decreased. Such a spontaneous alignment of polymer chains under the applied stress could support fiber formation during the electrospinning, especially during the production of parallel-aligned fibers [35]. The dynamic viscosity was almost constant during the measurement, meaning that the polycaprolactone–acetic acid–formic acid system behaves nearly as the Newtonian solution.

The hysteresis of shear stress was nearly suppressed for the composite precursor. It was probably caused by the hydroxyapatite nanoparticles, which prevented the polycaprolactone molecules from mutual arrangement. The dynamic viscosity η of the composite precursor exhibited shear thinning behavior. It is a typical behavior of suspensions when a phase separation takes place, making more space between the suspension constituents and hence decreasing the viscosity [36]. Such phase separation can cause some fluctuations in the electrospinning process resulting in places rich on the hydroxyapatite particles and places lacking the ceramic phase.

### 3.2. Electrospun Fibers Analysis

The obtained electrospun structures are shown in Figure 5, which also displays histograms of the measured fiber diameters. The actual average diameters and their standard deviations are listed in Table 3.

An obtained random electrospun structure from the pure polycaprolactone precursor (PCL-2) can be seen in Figure 5a. The structure contains minimum defects (drops, beads, etc.). The diameter of pure PCL fibers (sample PCL-2) was 172 ± 60 nm.

Figure 5b displays randomly oriented composite fibers (sample PCL/HA-2). With respect to pure PCL, the structure of PCL/HA-2 fibers contains more defects, and the fibers’ diameter is less homogenous. Small agglomerates of hydroxyapatite particles caused the formation of beads on the fibers. Hydroxyapatite particles penetrated through the walls of the fibers at some places and increased their surface roughness. The diameter of the composite fibers was comparable to the diameter of the PCL-2 fibers, but its standard deviation was nearly twice as large (171 ± 107 nm), which is apparent also from the wider distribution of diameters shown in the histogram inset of Figure 5b. Figure 5c shows random composite fibers after the plasma treatment (PCL/HA-3), whose structure is visually similar to the as-spun random composite fibers (PCL/HA-2). In addition, the diameter of the PCL/HA-3 fibers is comparable to the untreated fibers: 151 ± 75 nm.

Figure 5d shows parallel-organized composite fibers of PCL/HA-4. These fibers with an average diameter of 114 ± 60 nm were thinner than both types of randomly oriented fibers. Their diameter distribution was narrower than the distribution of the composite random nanofibers and shifted to the lower values compared to the PCL-2 nanofibers; see the histogram inset of Figure 5c. It was probably caused by the higher collector rotation (2000 rpm), which may induce tensile stress in the fibers, resulting in extreme stretching and thinning. Plasma-treated parallel composite fibers (PCL/HA-5) can be seen in Figure 5e. Their morphology is again similar to the as-spun fibers—the diameter of the PCL/HA-5 fibers is 127 ± 41 nm.

The typical diameter of the polycaprolacton electrospun fibers varies from 300 nm up to 1.3 μm, depending on the precursor concentration and electrospinning parameters [35]. Since not all the parameters can be always controlled, the lower diameter values of the prepared fibers do not suggest a deviation of the electrospinning process.

A pair comparison of Figure 5b with Figure 5c as well as Figure 5d with Figure 5e provides proof that the morphology of the fibers was not affected by the presence of plasma treatment.

Contact angle evaluation was performed on all the samples listed in Table 2 and for comparison, also on composite parallel-organized fibers before the plasma treatment (PCL/HA-4). The obtained contact angle values are summarized in Table 4. The contact angle of the pure polycaprolactone solid layer (PCL-1) was 75.5° ± 3.5°, whereas the contact angle of the polycaprolactone solid layer with hydroxyapatite particles PCL/HA-1 was 44.9° ± 1.7°. Such a difference is to be expected, as pure bulk hydroxyapatite is a hydrophilic material with a contact angle of approximately 10° [20].

As it is evident from Table 4, the fibrous structure has a negative influence on the hydrophilicity of the tested materials. The contact angle increased above 125° for all the fibrous (plasma untreated) samples. The increase is caused by microscopic surface roughness. This effect is known as the lotus effect. Briefly explained, microscopic surface features create an energetical barrier, preventing the liquid from spreading. Consequently, for the liquid, it is energetically more advantageous to create a liquid–gas interface than to flow over the microscopic surface barriers [37]. Interestingly enough, unlike for solid layers, the presence of HA particles did not lower the contact angle of our fibrous structures.

Plasma treatment resulted in rendering a completely wettable fibrous structure surface. The contact angle of the plasma-treated samples was unmeasurably small, as the drop soaked into the material immediately. To elucidate the submicroscopic mechanism behind the observed wetting improvement, samples were analyzed by means of XPS.

Figure 6 compares the XPS spectra of pure polycaprolactone fibers before and after the plasma treatment. The overall presence of bounded oxygen atoms was increased by approximately 2 percentage points. From deconvoluted O1s XPS spectra, it is apparent that argon plasma treatment enhanced the presence of C–O and H–O–C bonds on account of the C=O bond. However, it follows from Figure 7 that the hydroxyapatite powder remained untouched by the argon plasma treatment. The resistance of the hydroxyapatite to the argon plasma may be caused by the low energy of used discharge plasma and high bonding energies on the hydroxyapatite particles.

A similar type of argon plasma treatment mediated by dielectric barrier discharge in [30] was not able to render the PCL surface completely wettable. In [30], the contact angle was reduced from the initial 75° to 52°, at an energy density of 2.5 J/cm^2^. The reason is twofold. First, in our case, PCL samples were treated at a substantially higher power density of approximately 17 J/cm^2^. Second, the treatment in [30] (and similarly in [31]) was done at low pressure of 5 kPa, and special attention had been paid to samples degasifying from ambient gas residuals. In our case, the intended introduction of polar functional groups to the PCL surface relied on the dissociative interactions of excited argon species with adsorbed water and other oxygen-containing molecules. It is known from atmospheric pressure Ar plasma jet research that chemically reactive H, N, O, and OH radicals are formed in large quantities after the Ar jet nozzle exits to the surrounding environment of humid air [38]. The complex cascade of secondary and tertiary reactions that follows leads to the further formation of ‘long-living’ species of H_2_, O_2_, O_3_, H_2_O_2_, NO_2_, N_2_O, HNO_2_, and HNO_3_. Adsorbed moieties of ambient air on the PCL surface serve in the similar manner as ambient humid air in the plasma jet situation. Nevertheless, the pool of these potentially oxidative species is limited. Thus, even at higher plasma power inputs, the PCL substrate does not suffer by excessive oxidative degradation and a loss of its structural integrity. The difference in achieved surface modification is also apparent from the mutual pair comparison of XPS spectra. In [30] and [31], the formation of a new carbonyl C=O bond was clearly noticeable. In our case, the amount of already present surface carbonyl was reduced.

Taking advantage of the comprehensive list of chemical reactions summarized in [38], the following reactions of energized Ar species are expected to play a key role in the initial dissociative step:

metastables:

Ar(^4^S[^3^P_2_]) + O_2_ → 2O + Ar

Ar(^4^S[^3^P_2_]) + H_2_O → H + OH + Ar

excimers:

Ar_2_* + O_2_ → 2O + 2Ar

Ar_2_* + H_2_O → H + OH + 2Ar

ions:

Ar^+^_2_ + H_2_O → ArH^+^ + OH + Ar

In addition to these, direct electron impact dissociation reactions should be considered as well, owing to the close proximity of the treated PCL surface to the generated plasma (0.1 mm).

### 3.3. Biological Tests

The cell proliferation was evaluated every 24 h for 3 days. Figure 8 shows photos of the sample surfaces after 30 min and 72 h of testing. The difference between the cell colony after 30 min and after 72 h on the control sample (0) is shown in Figure 8a,b. It demonstrates the standard behavior of proliferating cells without any structural or composition effect. After 72 h, a non-continuous cell layer was created on the blank surface. The results after 72 h of cultivation on the surfaces of PCL and PCL/HA samples listed in Table 2 are shown in the Figure 8c–h. It is evident from Figure 8c,d that the solid PCL layer (PCL-1) did not initiate the cell proliferation contrary to its fibrous structure (PCL-2) counterpart. On the PCL-2 fibers, a cell layer comparable to the control sample was observed, which clearly indicates the positive effect of the fibrous microstructure on cell proliferation. However, the cell layer was not completely continuous. A similar character of the grown layer was observed on solid composite PCL/HA surfaces (Figure 8e) after 72 h. Unlike the pure PCL continuous layer (Figure 8c), the PCL/HA continuous composite layer (Figure 8e) enhanced cell proliferation, confirming the positive effect of HA particles. The continuous layer of the cells was formed on all fibrous PCL/HA composites with or without the plasma treatment (see Figure 8f,g,h). This is contrary to Cao et al. [39], who claimed the negative response of the cells to the large surface area to volume ratio of the polycaprolactone fibers. Good proliferation on plasma-treated samples proved that there are no undesired biocidal moieties introduced to the treated surface by plasma.

A continuous cell layer is considered as a successful indication of the bioactive properties of the tested material [40]. Hydroxyapatite particles occurred as a support to the cell proliferation on all PCL/HA samples. For a continuous polycaprolactone layer with hydroxyapatite particles, some positive effect of higher wettability may contribute as well. The ceramic particles would provide preferable places for cell adhesion and further proliferation.

Since the continuous cell layer had developed on both untreated and plasma-treated composite fibers, the effect of the surface plasma treatment on the sample bioactivity was not perspicuous. In order to detect any positive plasma effect on cell behavior, the proliferation rate was evaluated. The proliferation rate of the cells is plotted as a ratio of the specific growth rate (SGR) on the sample and on the control sample in Figure 9. A line representing the control has a value of 1 and is considered as a waterline; values above the waterline are considered as a positive result and vice versa.

The cell proliferation on the PCL-1 and PCL/HA -1 (the solid layers) samples was slower over 72 h than on the control sample. The low surface area of the continuous layer and hydrophobic nature of the polycaprolactone resulted in a worse proliferation rate evolution. Nevertheless, the proliferation rate of PCL/HA-1 was the highest during the first 24 h of cultivation. The better performance of PCL/HA-1 in comparison to PCL-1 can be explained by the presence of the hydroxyapatite nanoparticles increasing the hydrophilicity of the sample. During the first 24 h of the test, the cells invariably preferred occupying the places with higher concentration of the hydroxyapatite. When all the favored places were occupied, the cells’ activity slowed considerably. A similar trend occurred on fibrous plasma untreated samples (PCL-2 and PCL/HA-2), but the deceleration of the proliferation process came later (after 48 h) when the specific growth rate achieved its maximum. It was 10% higher than the control sample. At this point, the cells have already occupied all the favorable places such as fiber crossings and hydroxyapatite conglomerates (in the case of the PCL/HA-2 sample). Once occupied, there were no further positive stimuli available for the activity of the cells.

Both plasma-treated samples (PCL/HA-3, PCL/HA-5) exhibited the highest bioactivity with respect to the measured SGR compared with the untreated one, as can be seen in Figure 9. This agrees well with the former observations of [25] and [26] where the positive effect of plasma treatment was reported. The higher hydrophilicity of the whole plasma-treated structure induced the cell adhesion and subsequent proliferation [37]. The maximum SGR was reached after 48 h of cultivation and was by 30% higher than the control sample. Afterwards, the proliferation slowed down. It can be explained by the faster formation of the continuous cell layer itself. The cell proliferation slowed when the surface of the plasma-treated sample was fully covered by the cells. Further proliferation of the cells was inhibited by the lack of available space.

No effect on the specific growth rate of the cell culture of the fibers parallelly oriented was proven despite some reported studies [30]. Neither was there any strong influence of parallel-organized fibers (sample PCL/HA-5) on the shape of the cells. Roughly 50% of the cells showed a prolonged shape in the direction of the fiber alignment (PCL/HA-5); therefore, we cannot consider it as a thoroughly proven effect. The possible influence of the alignment of the fibers on the cell activity will have to be further studied.

## 4. Conclusions

The influence of the polycaprolactone and polycaprolactone/hydroxyapatite materials’ morphology and their surface modification on cell behavior was tested. It was confirmed that the topology of the electrospun fibrous structure has a positive influence on cell activity. In addition, the synergic effect of the hydroxyapatite particles incorporated into the fibrous structure and the surface plasma treatment was described. The results showed that the bioactivity offered by electrospun polymer/ceramic composites can be boosted effectively by a proper administration of inert gas plasma treatment/activation. For tissue engineering materials, this effect may contribute to a much faster adoption of transplanted tissue.

## Figures and Tables

**Figure 1 polymers-12-01877-f001:**
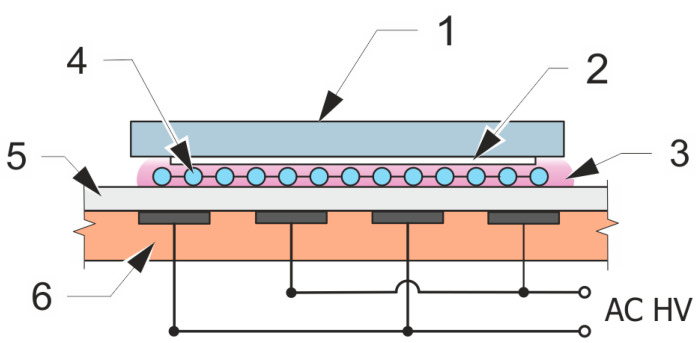
Scheme of plasma treatment set-up: (1) PP textile substrate; (2) treated PCL sample; (3) Ar discharge plasma; (4) spacer from PET mesh; (5) Al_2_O_3_ electrode; (6) electrode cooling medium.

**Figure 2 polymers-12-01877-f002:**
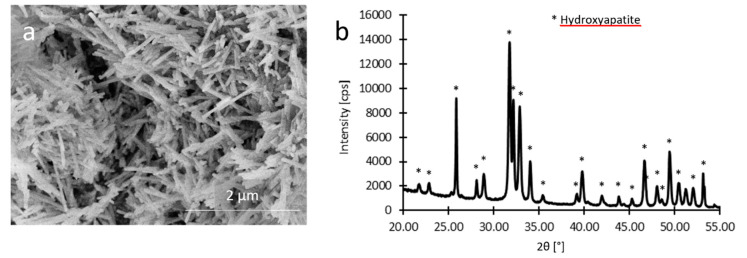
(**a**) SEM image of hydroxyapatite nanoparticles after the hydrothermal treatment, (**b**) XRD spectrum of synthetized hydroxyapatite particles.

**Figure 3 polymers-12-01877-f003:**
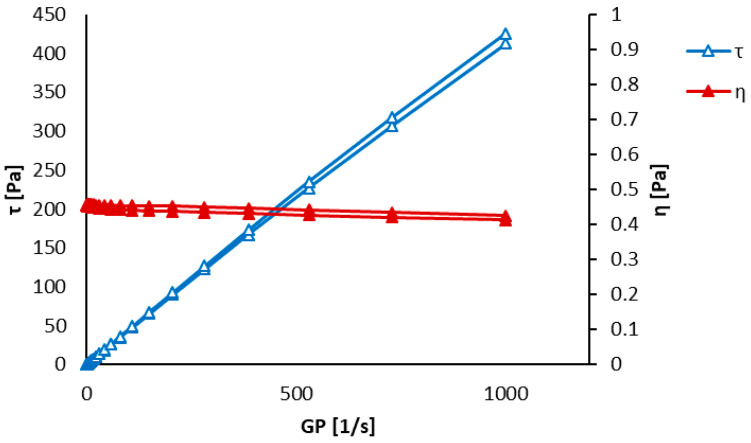
Dynamic viscosity measurement of a clear polycaprolactone precursor after 5 h of the preparation.

**Figure 4 polymers-12-01877-f004:**
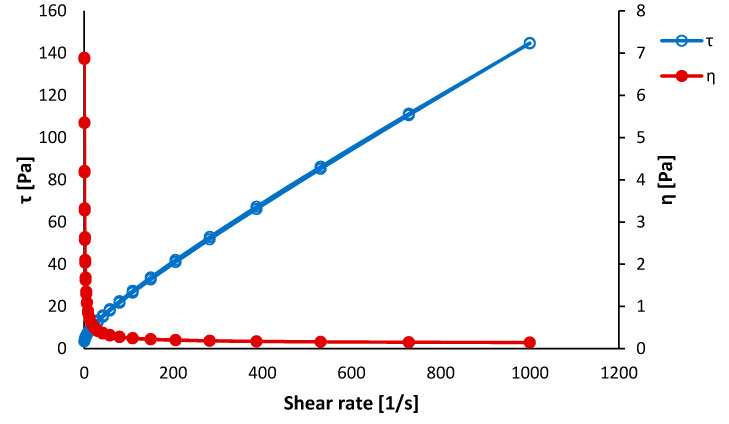
Dynamic viscosity measurement of a polycaprolactone/hydroxyapatite precursor after 5 h of the preparation.

**Figure 5 polymers-12-01877-f005:**
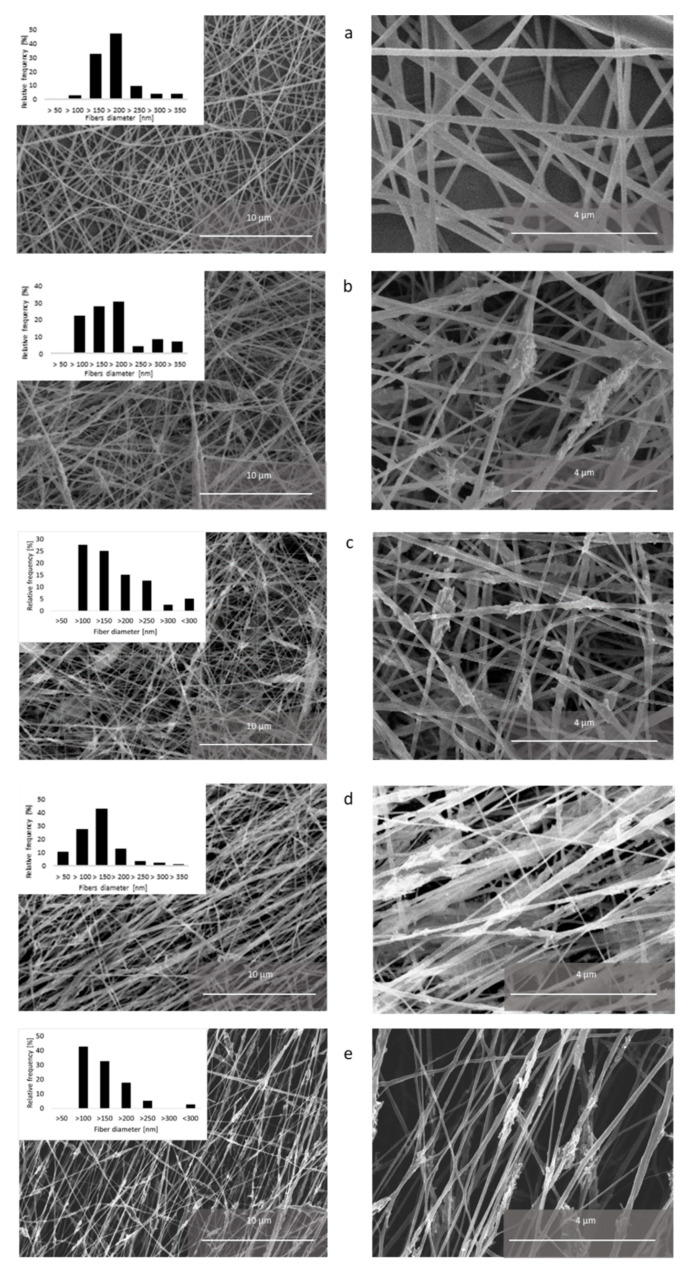
Electrospun fibers (**a**) PCL-2 (random polycaprolactone fibers), (**b**) PCL/HA-2 (random composite fibers), (**c**) PCL/HA-3 (random composite fibers after plasma treatment), (**d**) PCL/HA-4 (as-spun parallel composite fibers), and (**e**) PCL/HA-5 (parallel composite fibers after plasma treatment).

**Figure 6 polymers-12-01877-f006:**
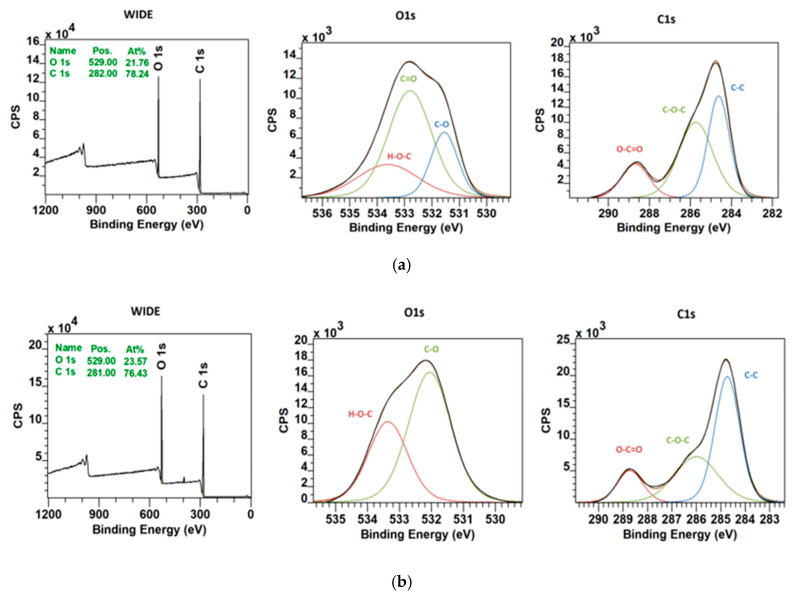
XPS analysis of the sample PCL-2 (**a**) before and (**b**) after the plasma treatment.

**Figure 7 polymers-12-01877-f007:**
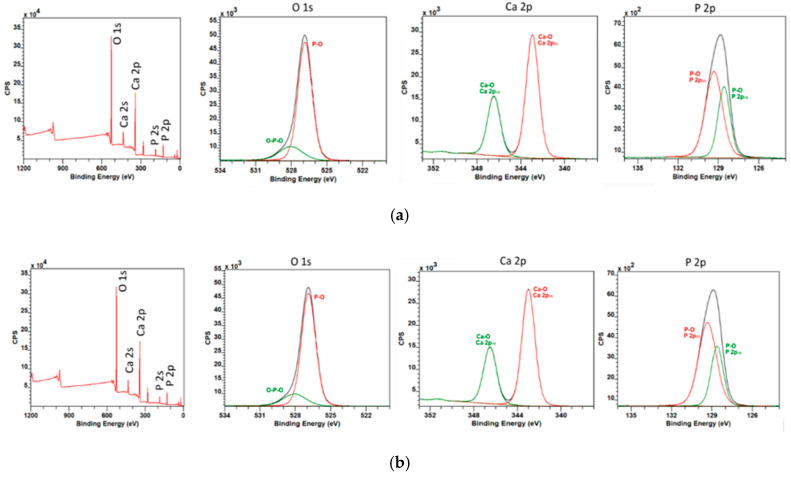
XPS analysis of the hydroxyapatite particles (**a**) before and (**b**) after the plasma treatment.

**Figure 8 polymers-12-01877-f008:**
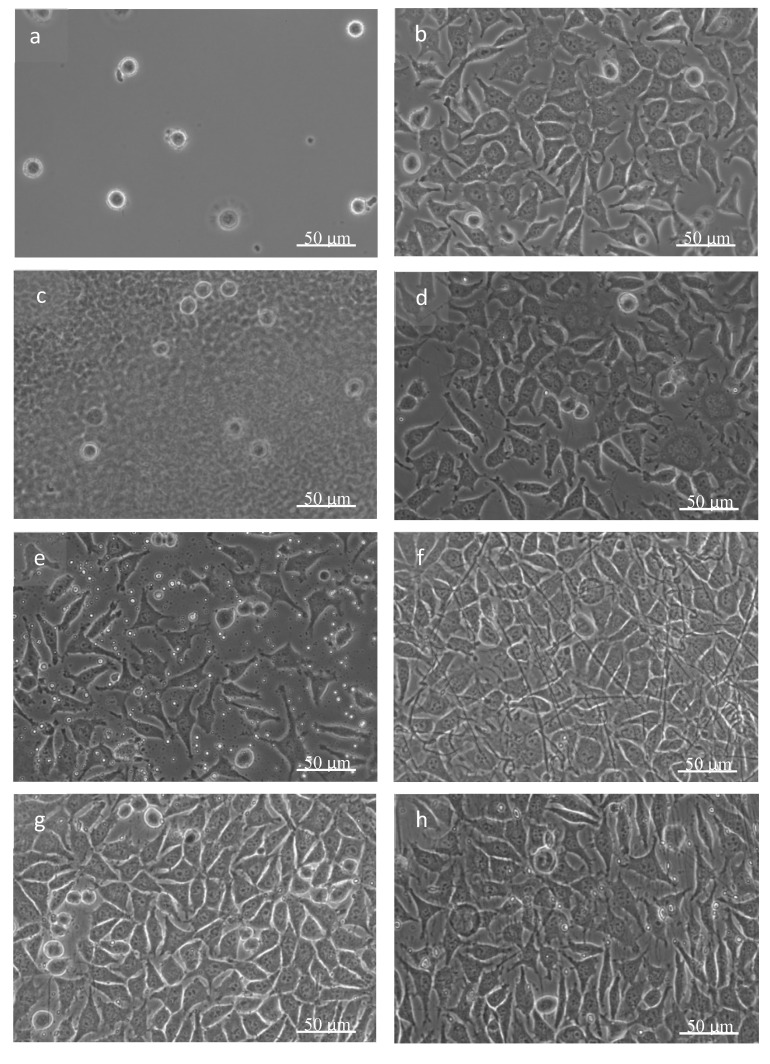
Photo of the cultivated cells on a sample surface, sample (**a,b**) 0 after 30 min and 72 h respectively, (**c**) PCL-1, (**d**) PCL-2, (**e**) PCL/HA-1, (**f**) PCL/HA-2, (**g**) PCL/HA-3, and (**h**) PCL/HA-5, all after 72 h.

**Figure 9 polymers-12-01877-f009:**
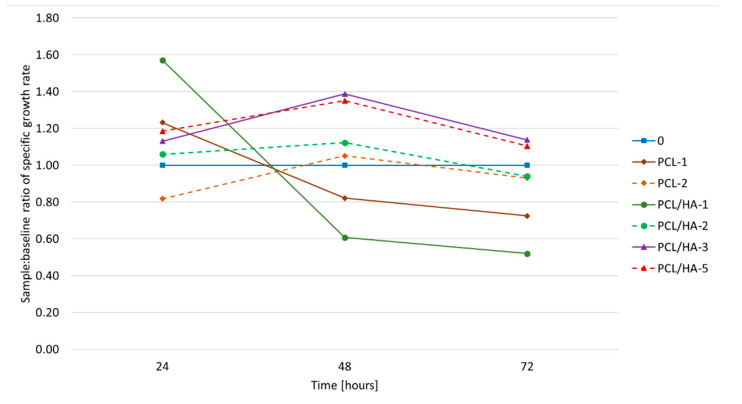
Specific growth rate ratio of the samples and the control.

**Table 1 polymers-12-01877-t001:** Electrospinning parameters.

Accelerating Voltage (kV)	Feeding Rate (µL/min)	Emitter–Collector Distance (mm)	Temperature (°C)
20	28	100	23

**Table 2 polymers-12-01877-t002:** Specification of the samples prepared via electrospinning. HA: hydroxyapatite, PCL: polycaprolactone.

Sample	Material	Morphology	Plasma Treatment
0 (baseline)	Clear glass	-	No
PCL-1	Polycaprolactone	Continuous layer	No
PCL-2	Polycaprolactone	Randomly organized fibers	No
PCL/HA-1	Polycaprolactone + hydroxyapatite particles	Continuous layer	No
PCL/HA-2	Polycaprolactone + hydroxyapatite particles	Randomly organized fibers	No
PCL/HA-3	Polycaprolactone + hydroxyapatite particles	Randomly organized fibers	Yes
PCL/HA-4	Polycaprolactone + hydroxyapatite particles	Parallel organized fibers	No
PCL/HA-5	Polycaprolactone + hydroxyapatite particles	Parallel organized fibers	Yes

**Table 3 polymers-12-01877-t003:** Diameter of the electrospun fibers.

Sample	Diameter [nm]
PCL-2	172 ± 60
PCL/HA-2	171 ± 107
PCL/HA-3	151 ± 75
PCL/HA-4	114 ± 60
PCL/HA-5	127 ± 41

**Table 4 polymers-12-01877-t004:** Results of contact angle evaluation.

Sample	Contact Angle [°]
PCL-1	75.5 ± 3.5
PCL-2	127.2 ± 4.1
PCL/HA-1	44.9 ± 1.7
PCL/HA-2	128.3 ± 3.6
PCL/HA-3	complete wetting
PCL/HA-4	125.3 ± 5.1
PCL/HA-5	complete wetting

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
