# Peer review of "Influence of Hydroxyapatite Nanoparticles and Surface Plasma Treatment on Bioactivity of Polycaprolactone Nanofibers"

_polymers, 2020, doi:10.3390/polym12091877_

Round 1

Reviewer 1 Report

Authors prepared polycaprolactone nano-sized fibres with hydroxyapatite by electrospinning method. Composites morphology (random or parallelly oriented fibers vs. continuous layer) /  composition as well as the influence of  low temperature argon discharge plasma treatment on cell proliferation was then tested. 

Polycaprolactone is a broadly used biocompatible polymer that can be easily processed by various methods, including electrospinning. Also PCL composites with hydroxyapatite hasve been very well known since decades. The reviewed work shows however certain  scientific novelty by linking multi-phase nanofibrous systems with plasma treatment.

Although the manuscript is well-structured, some points need more discussion:

  • please provide information on the composition of synthetized hydroxyapatite - is that a stoichiometric composition with Ca/P ratio of 1.67?
  • The influence of plasma treatment -that is the key novelty of this work -  is poorly discussed.  Please expand considerably this section with discussion on the mechanism, role of HAp, what happens in the interfacial PCL/HAp regions, and how the composite structure is affected towards biological properties?
  • Fig. 8 - is it possible to have it in colour? It would help much to analyse the morphology.
  • Conclusions are not the abstract - please re-write this section.

Besides, in lines 195-196 there is a sentence "This research was also financially supported by the Ministry of 196 Education, Youth and Sports of the Czech Republic under the project LTT18013"   which is then repeated under  "Funding".

Reviewer 2 Report

The article presents an extensive study of the effect of insertion of hydroxyapatite nanoparticles and atmospheric non-thermal plasma treatment on biactivity of polycaprolactone nanofibers. In general, the article is well written and presents relevant results on the subject, in special on the effect of plasma treatment in cell proliferation, thus being of interest to readers of Polymers journal. 

It is recommended that authors review English language, as well as several typos. The formation of chemical formulas should also be revised, since the numerical indices are not underwritten. 

In addition, to improve the quality of the article, some notes are made:

(1) Please change the form of quotation "and col." for "et al.", "and col." it is not commonly used in the literature.

(2) In Table 2, please verify if it is correct the indications of plasma treatment samples. It is not in consonance with the results presented.

(3) In lines 195 and 196 please remove the paragraph: "This research was also financially supported by the Ministry of 196 Education, Youth and Sports of the Czech Republic under the project LTT18013.". This is mentioned in the topic Funding.

(4) Attention to the use of a comma in the decimals of figures and tables.

(5) Question: Why didn't the authors investigate the effect of plasma on PLC samples? It is interesting to highlight the effect of plasma in samples that do not contain hydroxyapatite.
